# Assessing Postural Stability Using Coupling Strengths between Center of Pressure and Its Ground Reaction Force Components

Jia-Li Sung [1], Lan-Yuen Guo [2], Chin-Hsuan Liu [3,4], Posen Lee [4], Chen-Wen Yen [1,5,6,*] and Lih-Jiun Liaw [5,6,7,8,*]

[1] Department of Mechanical and Electro-Mechanical Engineering, National Sun Yat-Sen University, Kaohsiung 80424, Taiwan; d033020003@student.nsysu.edu.tw

[2] Department of Sports Medicine, Kaohsiung Medical University, Kaohsiung 80708, Taiwan; yuen@kmu.edu.tw

[3] Department of Occupational Therapy, Kaohsiung Municipal Kai-Syuan Psychiatric Hospital, Kaohsiung 80276, Taiwan; isu6a394@cloud.isu.edu.tw

[4] Department of Occupational Therapy, I-Shou University, Kaohsiung 82245, Taiwan; posenlee@isu.edu.tw

[5] Department of Physical Therapy, College of Health Science, Kaohsiung Medical University, Kaohsiung 80708, Taiwan

[6] Neuroscience Research Center, Kaohsiung Medical University, Kaohsiung 80708, Taiwan

[7] Department of Physical Medicine and Rehabilitation, Kaohsiung Medical University, Kaohsiung 80708, Taiwan

[8] Department of Medical Research, Kaohsiung Medical University, Kaohsiung 80708, Taiwan

\* Correspondence: vincen@mail.nsysu.edu.tw (C.-W.Y.); lijili@kmu.edu.tw (L.-J.L.);
Tel.: +886-7-5254232 (C.-W.Y.); +886-7-3121101 (ext. 2663) (L.-J.L.)

**Abstract:** The center of pressure (COP), which is defined as the point at which the resultant ground reaction force (GRF) is applied on a body, provides valuable information for postural stability assessment. This is because the fundamental goal of balance control is to regulate the center of mass (COM) of the human body by adaptively changing the position of the COP. By using Newtonian mechanics to develop two equations that relate the two-dimensional COP coordinates to the GRF components, one can easily determine the location of the COP using a force plate. An important property of these two equations is that for a given COP position, there exists an infinite number of GRF component combinations that can satisfy these two equations. However, the manner in which a postural control system deals with such redundancy is still unclear. To address this redundancy problem, we introduce four postural stability features by quantifying the coupling strengths between the COP coordinates and their GRF components. Experiments involving younger (18–24 years old) and older (65–73 years) participants were conducted. The efficacy of the proposed features was demonstrated by comparing the differences between variants of each feature for each age group (18–24 and 65–73 years). The results demonstrated that the coupling strengths between the anterior–posterior (AP) direction coordinate of the COP and its GRF components for the older group were significantly higher than those of the younger group. These experimental results suggest that (1) the balance control system of the older group is more constrained than that of the younger group in coordinating the GRF components and (2) the proposed features are more sensitive to age variations than one of the most reliable and accurate conventional COP features. The best testing classification accuracy achieved by the proposed features was 0.883, whereas the testing classification accuracy achieved by the most accurate conventional COP feature was 0.777. Finally, by investigating the interactions between the COP and its GRF components using the proposed features, we found that that the AP component of the GRF of younger people plays a more active role in balance control than that of the GRF of older people. Based on these findings, it is believed that the proposed features can be

used as a set of stability measures to assess the effects on posture stability from various health-related conditions such as aging and fall risk.

**Keywords:** balance control; quiet standing; ground reaction force; force plates

## 1. Introduction

Postural stability deterioration is a risk factor in older people falling [1–4]. Because falling is a major cause of morbidity and mortality in -older people [4], improving postural stability is important. Therefore, simple and effective postural stability assessment methods are highly valuable and should be developed.

Postural stability is closely related to the interaction between weight force and ground reaction force (GRF). The weight force acts at the center of mass (COM) of the human body, and the GRF acts at the center of pressure (COP), which is the point at which the resultant GRF is applied below the body. To maintain postural stability, the vertical projection of the COM must remain within the polygon of support. To this end, the postural control system must generate stabilizing moments to regulate the COM by adaptively changing the position of the COP [5,6]. Consequently, many postural stability measures have been developed using the trajectories of the COM and COP [7–13]. Such development has been particularly active for the COP because it can be easily measured using force plates. In fact, review papers have indicated that COP features have been employed in approximately 60% of published studies on postural control [14,15]. In addition, COP features have also been extensively employed to assess the effects on posture stability from various health-related conditions, such as aging [13,16], peripheral neuropathy [17], musculoskeletal disorders [18], stroke [19,20], spinal cord injury [21], concussion [22], cancer [23], frailty syndrome [24], symptomatic degenerative lumbar disease [25], Parkinson's disease [26,27], multiple sclerosis [28,29], and high fall risk [30]. Interested readers are referred to the review papers by Visser et al. [31] and Chaudhry et al. [32] for the background and utility of posturography.

As detailed in the Methods section, the number of those GRF components (measured using force plates) that are associated with the COP is higher than the degrees of freedom (DOFs) of the COP. Therefore, there necessarily exists an infinite number of GRF component combinations that can satisfy a given COP position requirement. This phenomenon is similar to the motor redundancy problem formulated by Bernstein [33]. As an important problem in motor control, motor redundancy occurs when the number of motor control task constraints is lower than the number of control variables of the motor control system. Therefore, the motor control system must incorporate strategies for coordinating these redundant DOFs. Possible strategies include: first, simplifying the complexity of the motor control task by synchronizing the DOFs of the motor control system [33,34]; second, using the redundant DOFs to optimize the performance of the control task [35], and; third, assisting the operation of secondary tasks [36]. However, to the best of our knowledge, the redundancy problem has never been systematically studied for the COP.

To investigate the potential utility of such redundancy to improve postural stability, instead of relying only on information extracted from the COP, we develop new postural stability features by quantifying the coupling strengths between the COP and its GRF components. In addition to developing new postural stability measures, we hope that the experimental results will elucidate how our postural control system utilizes the redundant GRF components to assist balance control.

## 2. Methods

### 2.1. Proposed Features

Figure 1 illustrates a typical force plate that is used in this study. Force transducers are placed at the four corners of the force plate. In Figure 1, the $x$-, $y$-, and $z$-axes correspond to the medial-lateral (ML) and anterior-posterior (AP), and vertical directions, respectively, and the origin of the coordinate system represents the center of the force plate. Note that the arrows associated with the three axes point to the positive direction of the corresponding axes.

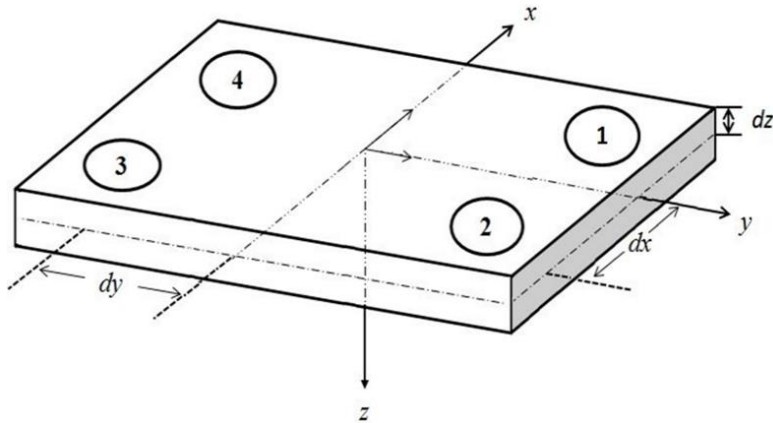

**Figure 1.** Force plate, its coordinate system, its three geometrical parameters, and locations of the four force sensors.

With $F_i$ denoting the GRF component at the $i$th corner of the force plate, and $f_x$, $f_y$, and $f_z$ representing the GRF components in the ML, AP, and vertical directions, respectively, the resultant GRF can be expressed by the following vector $f$:

$$f = f_x i + f_y j + f_z k = F_1 + F_2 + F_3 + F_4 \tag{1}$$

where $i$, $j$, and $k$ are unit vectors along the $x$-, $y$-, and $z$-axes, respectively. With $F_{iz}$ denoting the vertical component of $F_i$, the vertical component of the resultant GRF $f$ can be written as:

$$f_z = F_{1z} + F_{2z} + F_{3z} + F_{4z} \tag{2}$$

Similarly, with $F_{ix}$ and $F_{iy}$ denoting the ML and AP components of $F_i$, respectively, the ML and AP components of the resultant GRF $f$ can be represented as:

$$f_x = F_{1x} + F_{2x} + F_{3x} + F_{4x} \tag{3}$$

$$f_y = F_{1y} + F_{2y} + F_{3y} + F_{4y} \tag{4}$$

To determine the coordinates of the COP, we must know three geometrical parameters of the force plate: $d_x$, $d_y$, and $d_z$. As shown in Figure 1, $d_x$ and $d_y$ represent the distances along the $x$-axis and $y$-axis, respectively, from the coordinate axes to the force sensors, and $d_z$ is the distance along the $z$-axis from the origin of the coordinate system to the support surface of the force plate. With $d_x$, $d_y$, and $d_z$ known, the following equations can be derived using Newtonian mechanics to determine the coordinates of the COP.

$$COP_x = \frac{d_x F_{xz} + d_z f_x}{f_z} \tag{5}$$

$$COP_y = \frac{d_y F_{yz} + d_z f_y}{f_z} \tag{6}$$

where

$$F_{xz} = F_{1z} + F_{4z} - F_{2z} - F_{3z} \qquad (7)$$

$$F_{yz} = F_{1z} + F_{2z} - F_{3z} - F_{4z} \qquad (8)$$

Equation (7) demonstrates how the four vertical GRF components are combined to influence the motion of $COP_x$ which is the ML direction coordinate of COP. Similarly, Equation (8) shows how the four vertical GRF components are combined to change the motion of $COP_y$ which is the AP direction coordinate of the COP.

During quiet standing, the vertical direction component of the inertia force is considerably weaker than the weight force, and therefore, we assume that $f_z$ equals the weight force during quiet standing. Consequently, with $M$ as the mass of the test subject and $g$ as the acceleration due to gravity, we have:

$$f_z = F_{1z} + F_{2z} + F_{3z} + F_{4z} \approx Mg \qquad (9)$$

Under the assumption of a constant $f_z$, Equations (5) and (6) can be rewritten as:

$$COP_x = c_{xz}F_{xz} + c_{zz}f_x \qquad (10)$$

$$COP_y = c_{yz}F_{yz} + c_{zz}f_y \qquad (11)$$

where $c_{xz} = d_x/Mg$, $c_{zz} = d_z/Mg$, and $c_{yz} = d_y/Mg$. Equation (10) shows that $COP_x$ is determined by its vertical GRF component $F_{xz}$ and its medial-lateral horizontal GRF component $f_x$. Similarly, Equation (11) indicates that $COP_y$ is determined by its vertical GRF component $F_{yz}$ and its anterior-posterior horizontal GRF component $f_y$.

For a given COP location, Equations (9)–(11) represent an underdetermined system because the number of constraints is lower than the number of variables that can be used to satisfy the constraints. Consequently, there exists an infinite number of combinations of $F_{1z}$, $F_{2z}$, $F_{3z}$, $F_{4z}$, $f_x$, and $f_y$ that can satisfy Equations (9)–(11). However, the manner in which the human postural control system manages this redundancy problem is unclear.

A possible approach to partially resolve this redundancy problem associated with Equation (11) is to choose $f_y$ to be a constant multiple of $F_{yz}$. Under such a constraint, both $F_{yz}$ and $f_y$ will be perfectly correlated with $COP_y$. To evaluate the validity of this possibility, we formulated our first proposed feature to quantify the coupling strength between $COP_y$ and $F_{yz}$. Specifically, because such a coupling relationship may be time-varying, the time responses of $COP_y$ and $F_{yz}$ are divided into time subintervals. Via a trial-and-error process, the length of these non-overlapping subintervals was chosen to be 0.125 s. For each time subinterval, the Pearson product-moment correlation coefficient is used to quantify the coupling strength. The first proposed feature, denoted as $\alpha_{AP}$, is defined as the mean of these correlation coefficient values. Similarly, we formulated our second proposed feature $\beta_{AP}$ to characterize the coupling strength between $COP_y$ and $f_y$.

To investigate the redundancy problem associated with Equation (10), we introduce two additional features that are similar to $\alpha_{AP}$ and $\beta_{AP}$. Denoted as $\alpha_{ML}$ and $\beta_{ML}$, these two additional features characterize the coupling strength between $COP_x$ and $F_{xz}$ and the coupling strength between $COP_x$ and $f_x$, respectively. In summary, $\alpha_{AP}$ characterizes the coupling strength between $COP_y$ and its GRF component in the vertical direction $F_{yz}$, whereas $\beta_{AP}$ characterizes the coupling strength between $COP_y$ and its GRF component in the anterior-posterior horizontal direction $f_y$. The remaining two proposed features, $\alpha_{ML}$, and $\beta_{ML}$, for the ML direction are defined in a similar manner.

## 2.2. Experimental Procedure

The volunteers participating in this study belonged to two age groups: a younger age group (20.1 ± 1.29 years, range: 18–24 years; BMI: 22.5 ± 3.21 kg/m$^2$) and an older age group (68.7 ± 2.96 years, range: 65–73 years; BMI: 23.9 ± 4.13 kg/m$^2$). Each group comprised of 10 healthy men and 10 healthy

women. Based on a self-report and a physical examination, we determined that none of the subjects had a pathological condition that would compromise their postural performance. The experimental procedures were approved by the Institutional Review Board of the Kaohsiung Medical University Chung-Ho Memorial Hospital, Kaohsiung, Taiwan.

All subjects were tested in two experiment sessions per day for two days. Each session included three 80-s trials. During the first 40 s of the trials, the subjects were asked to look straight ahead at a visual reference and stand still (with their arms at the side) in a comfortable stance near the center of the force plate. Subsequently, under the same test conditions, the subjects were asked to close their eyes for the remaining 40 s of the trial. Approximately 1 min and 5 min of rest were given between each trial and each session, respectively. Data collected between the 5th and 35th second of the eyes-open trials were used in this study.

The measurement system comprised a force plate (9286AA, KISTLER Instrumente AG, Winterthur, Switzerland) connected to a PC-based data acquisition system. The force plate measurements were sampled at 512 Hz with a 14-bit analog-to-digital data acquisition card (USB-6009, National Instruments) connected to a desktop PC. The data processing software was a custom-developed program written in LabVIEW (National Instruments, Austin, TX, USA). The signals were filtered by a zero-phase sixth-order low-pass Butterworth filter with a cutoff frequency of 5 Hz. Finally, for quantifying the coupling strengths between signals, the length of the time subintervals was chosen to be 0.125 s.

## 2.3. Data Analyses

The data were analyzed to assess the differences between the younger and older groups by using the tested features. In addition to the proposed features, the tested features included COP velocity features. Conventionally, the COP velocity is characterized by its mean velocity ($MV$), its mean velocity in the ML direction ($MV_{ML}$), and its mean velocity in the AP direction ($MV_{AP}$). This study chose $MV_{AP}$ and $MV_{ML}$ as the benchmark reference features because many studies have shown that they are reliable and informative COP measures for postural steadiness [37–41]. This is particularly true for $MV_{AP}$, which has been considered the most sensitive measure for assessing postural balance performance [37,42,43].

To compare the efficacy of the proposed features, we first used independent two-sided *t*-tests to compare the means of the tested features of the younger and older age groups. A statistically significant difference was indicated if $p < 0.05$. The statistical results were obtained using the mean values of the tested features over twelve trials. The sample size for both age groups was 20. The results are summarized in Table 1.

**Table 1.** Statistical results for the tested features.

| Features | Groups | | *p*-Values |
|---|---|---|---|
| | Younger Group | Older Group | |
| $\alpha_{ML}$ | 0.995 ± 0.003 | 0.994 ± 0.005 | 0.635 |
| $\beta_{ML}$ | 0.387 ± 0.101 | 0.427 ± 0.118 | 0.256 |
| $MV_{ML}$ | 4.466 ± 1.055 | 4.233 ± 0.825 | 0.443 |
| $\alpha_{AP}$ | 0.982 ± 0.009 | 0.995 ± 0.003 | 6.62E–07 |
| $\beta_{AP}$ | 0.523 ± 0.056 | 0.677 ± 0.068 | 1.88E–09 |
| $MV_{AP}$ | 5.447 ± 0.973 | 7.278 ± 1.990 | 6.82E–04 |

Values of the second and third columns are mean ± standard deviation. Feature units: dimensionless, $\alpha_x$, $\beta_x$, $\alpha_y$, and $\beta_y$; mm/s, $MV_{AP}$ and $MV_{ML}$.

To further compare the efficacy of the tested features, we used each of the tested features to classify the two age groups. Because the generalization capability of a classifier depends strongly on the size of the training set, this study employed a simple three-trial averaging technique to increase the number of samples. Therefore, because we have 12 trial results (4 sessions × 3 measurements/session) for each test subject and because the number of possible combinations for selecting three objects from 12 objects

is 220, by using the average of three trial results as the sample to be classified, this three-trial averaging technique generated 220 samples for each subject. Consequently, this simple three-trial averaging technique increased the size of the data set from 240 samples (12 measures/person × 20 persons) to 4400 samples (220 measures/person × 20 persons) for both age groups. Finally, in addition to accuracy, we also computed the area under the curve (AUC) of the receiver operating characteristic (ROC) curve. The AUC is one of the most effective performance measures for binary classification problems [44].

To obtain the ROC curve for each of the tested features, we assigned the older age group to the positive class when the sample mean of the tested feature was higher in the older group than in the younger group. Otherwise, the older group was assigned to the negative class. When classifying any given sample data point, if its feature value was higher than the decision threshold, then the binary classifier assigned this data point to the positive class. Otherwise, this data point was classified as a negative class sample. If a positive class sample was correctly classified as positive, it was considered a true positive. Conversely, if a negative class sample was incorrectly classified as positive, it was considered a false positive. Apparently, false positive and true positive rates vary with the value of the decision threshold. By sweeping the threshold value from the lowest to the highest values of the tested feature, we generated the ROC curve by plotting the true positive rate as a function of the false positive rate. From the ROC curve, we determined the optimal value for the decision threshold that results in the highest classification accuracy.

To assess the effectiveness of the tested features, one can use the entire dataset to determine the ROC curve and the corresponding optimal decision threshold. However, with a limited number of participants, whether such results can be accurately generalized to the general population is questionable. To address this issue, this study used a generalization test procedure to assess the effectiveness of the tested features. Specifically, this generalization test procedure randomly selected 80% of the participants (8 men and 8 women for each age group) as the training subset and assigned the remaining 20% of the participants (2 men and 2 women for each age group) to the testing subset. For each of the tested features, this generalization procedure performed the following steps. First, based on the classification method described in the previous paragraph, this procedure used the training subset to determine the optimal decision threshold. Next, this decision threshold was used to classify the testing subset samples. With such a randomly divided training and testing subsets, this procedure was repeated 1000 times for each of the tested features. To characterize the effectiveness of a tested feature, the final step determined the average accuracy of these 1000 generalization test results.

One of the limitations of the three-trial averaging technique is that each of the participants needed to have at least three trial results. Therefore, the effectiveness of the proposed approach becomes unclear in dealing with subjects with less than three trial measurements. To address this problem, we repeated the generalization tests by revising the testing subset samples. Specifically, the decision thresholds were still determined by using training samples generated from the three-trial averaging technique. However, the testing subset used the single-trial measurements of the testing subset participants to characterize the effectiveness of the tested features. Therefore, the number of samples associated with each of the testing subset participants reduced from 220 (the number of combinations of selecting 3 trails from 12 trials) to 12 (each participant has 12 trial results). Note that by repeating this procedure 1000 times with randomly divided training and testing subsets, this method tested 48,000 samples (12 trials/participants * 4 testing subset participants/test * 1000 generalization tests) for each age group.

## 3. Results

The means of the tested features for the two age groups were compared. As indicated in Table 1, the three ML-direction tested features were all nonsignificant in the statistical analysis. By contrast, the mean values of the three AP-direction tested features of the older group were all significantly higher than those of the younger group.

To address the potential problem of limited dataset size, Table 2 summarizes the classification results of the training and testing subsets. As illustrated in the previous section, by randomly assigning

80% and 20% of the male and female participants to the training and testing subsets, respectively, the generalization test procedure was repeated 1000 times. Results reported in Table 2 are the averages of these 1000 generalization test results. Note the AUC results of the testing subsets are not included in Table 2. The reason is that the ROC curves for these three generalization test subsets need to be independently determined. As a result, the AUC results of the testing subsets can not characterize the generalization performance of the tested features.

**Table 2.** Classification results of the tested features.

| Features | Classification Accuracy Results | | | |
| --- | --- | --- | --- | --- |
| | Training Subsets | | Testing Subsets | |
| | | | 3-Trial Averaging Samples | Single-Trial Samples |
| | Accuracy | AUC | Accuracy | Accuracy |
| $\alpha_{ML}$ | 0.576 | 0.504 | 0.563 | 0.548 |
| $\beta_{ML}$ | 0.606 | 0.589 | 0.590 | 0.583 |
| $MV_{ML}$ | 0.593 | 0.545 | 0.593 | 0.578 |
| $\alpha_{AP}$ | 0.818 | 0.886 | 0.810 | 0.727 |
| $\beta_{AP}$ | 0.903 | 0.954 | 0.895 | 0.839 |
| $MV_{AP}$ | 0.739 | 0.777 | 0.718 | 0.682 |

Note that, for each of the tested features, Table 2 provides three sets of classification results which include results of the training subset, three-trial averaging sample testing subset, and the single-trail sample testing subset. To study the differences between these three sets of results, as a representative case, here we compared the results associated with feature $\beta_{AP}$. As shown in Table 2, for $\beta_{AP}$, the classification accuracy of the training subset, the three-trial averaging sample testing subset, and the single-trial sample testing subset were 0.903, 0.895 and, 0.839, respectively. As expected, the training subset had the best accuracy since the decision threshold was designed to minimize the classification error of the training subset samples. In comparison, by using such a decision threshold to classify the testing subset samples, the classification results were expected to be less accurate since the probability distribution function of the training subset was not identical to that of the testing subsets. Another important observation was that the classification accuracy of the training subset and the three-trial averaging sample testing subset were relatively close (0.903 versus 0.895). This suggested that, even with a relatively small number of training subset participants, $\beta_{AP}$ still performed very well in dealing with testing subset participants whose information was completely hidden from the classification rule development process.

For $\beta_{AP}$, the testing subset classification accuracy of the three-trial averaging samples was better than that of the single-trial samples (0.895 vs. 0.839). Such a difference was also expected since the decision threshold was determined by using three-trial averaging samples. Such a result also suggested that increasing the number of measurements could improve the effectiveness of $\beta_{AP}$ in differentiating the two age groups. Finally, although here we only explicitly discuss the results associated with $\beta_{AP}$, the comparative results of the remaining tested features were basically the same.

## 4. Discussion

The results listed in Table 1 for $MV_{AP}$ and $MV_{ML}$ agree with those of previous studies, indicating that $MV_{AP}$ is more effective than $MV_{ML}$ in assessing the postural unsteadiness differences between different age groups and between different health-related conditions [13,31].

The results listed in Table 2 also demonstrated that the AP-direction tested features were more sensitive in detecting aging effects than the ML-direction tested features were. The best results were obtained by the proposed feature $\beta_{AP}$ (with a training subset classification accuracy of 0.903 and AUC of 0.954). By comparison, the training subset classification accuracy and AUC achieved by the conventional COP feature $MV_{AP}$ were 0.739 and 0.777, respectively. Moreover, the classification

accuracy and AUC of the AP-direction features were higher than those of the ML-direction features. Thus, the following discussions focus only on AP-direction features. Note that the results of the discussions can be easily generalized to the ML-direction features. To interpret the results obtained by the proposed features, the remaining part of this section addresses the redundancy problem between $COP_y$ and its GRF components. As shown by Equation (11), there exists an infinite number of combinations of $F_{yz}$ and $f_y$ for a given $COP_y$. By studying the interactions between these three signals, we attempted to investigate how our balance control system deals with such a redundancy problem.

The $\alpha_{AP}$-characterized correlation between $COP_y$ and $F_{yz}$ was high because the mean values of the proposed feature $\alpha_{AP}$ (0.982 and 0.995 for the younger and older age groups, respectively) were very close to unity (Table 1). By comparison, the $\beta_{AP}$-characterized correlation between $COP_y$ and $f_y$ was weaker than that of the correlation between $COP_y$ and $F_{yz}$ because the values of $\beta_{AP}$ (0.523 and 0.677 for the younger and older age groups, respectively) were considerably lower than the values of $\alpha_{AP}$. These results suggested that $COP_y$ is largely determined by $F_{yz}$. This observation was also evinced by the signal energy of $F_{yz}$ being much higher than that of $f_y$. In this study, the average value of the energy of the $F_{yz}$ signal was approximately 1803 times that of the $f_y$ signal. Similar results were also observed in the ML direction. Therefore, we concluded that the COP was largely determined by the vertical GRF components during quiet standing; this finding explained why several previous studies on quiet standing have neglected the contributions of the horizontal GRF components to the COP [45–47].

Despite the dominant role of $F_{yz}$ in determining $COP_y$, the following three observations suggested that the influence of $f_y$ is not negligible. First, as indicated in Table 1, the $\alpha_{AP}$ value of the younger group was significantly lower than that of the older group ($p = 6.62 \times 10^{-7}$). This observation suggested that compared with that of the older group, the $f_y$ of the younger group played a more active role in determining $COP_y$.

Our second observation is based on Newton's second law of motion, which states that the acceleration of an object is proportional to the net force acting on the object. The second law implies the following:

$$COM_y \propto f_y \tag{12}$$

where $C\ddot{O}M_y$ is the acceleration of the AP component of the COM. If $f_y$ is highly coupled with $COP_y$, $f_y$ loses its ability to independently maneuver the AP component of the COM. This can impair postural stability because the fundamental goal of postural balance control is to maintain the vertical projection of the COM within the safe limits of the support surface. This could partially explain why the values of the proposed feature $\beta_{AP}$, unlike the values of $\alpha_{AP}$, were not very close to 1. In addition, the values of $\beta_{AP}$ summarized in Table 1 demonstrated that the correlation strength between $COP_y$ and $f_y$ for the older group was significantly higher than that for the younger group. Hence, because $f_y$ was more strongly coupled with $COP_y$, the $f_y$ force component of the older group was more constrained to maneuver the AP component of the COM. As a result, postural stability can be undermined.

The third observation was related to the effect of $f_y$ on the vertical free moment (FM). As a frictional torque, FM can be computed from the following equation [48,49]:

$$FM = M_z - COP_x f_y + COP_y f_x \tag{13}$$

where $M_z$ is the vertical moment about the force plate center. Note that including $M_z$, every signal that appeared on the right-hand side of Equation (13) could be directly measured using a force plate. Therefore, the FM is an additional postural variable that can be easily obtained using only a force plate.

Similar to Equations (10) and (11), which indicated that moving the COP to the desired position required the coordination of different GRF components, Equation (13) demonstrated that $M_z$, $f_x$, and $f_y$ must coordinate with each other to limit the magnitude of FM to prevent the body from rotating excessively in the vertical direction. The capability of $f_y$ in freely attending such a coordination task was weakened when $f_y$ was highly correlated with $COP_y$. As demonstrated by the results of $\beta_{AP}$ in

Table 1, this weakening effect was more prominent in the older group than in the younger group because $\beta_{AP}$ characterizes the coupling strength between $COP_y$ and $f_y$.

In summary, in studying the redundancy problem associated with Equation (11), we found that the coupling strengths between $COP_y$, $F_{yz}$, and $f_y$ of the younger group were lower than those of the older group. As a result, the younger group exhibited higher flexibility in coordinating $F_{yz}$ and $f_y$ when forming a given $COP_y$. In addition, by more loosely coupling $f_y$ with $COP_y$, the $f_y$ force component of the younger group could play a more active role than that of the older age group in managing the AP component of the COM and the vertical FM. This relatively more active role of $f_y$ in the younger age group could partially explain why the younger group's posture balance was more stable than that of the older group.

In this study, the efficacy of the proposed approach was demonstrated using comparative quiet standing experiments involving participants of two age groups. By quantifying the coupling strengths between COP and its GRF components, the proposed features tried to characterize the complex interplay of biomechanics and neuromuscular control during the balancing process. As discussed in Section 1, conventional COP features were employed to characterize the effects of many health-related conditions on posture balance [13,16–30]. We intend to further test the efficacy of the proposed features in future similar studies. In specific, by performing longitudinal studies to observe how the values of the proposed features vary with the disease severity, the proposed approach can become an effective tool for monitoring the progress of diseases.

Perhaps one of the most important potential applications of the proposed approach is to identify older people with high fall risk. As addressed in a review paper, conventional COP features can be used to discriminate fallers from non-fallers [30]. Based on the favorable comparative results obtained in this work, it seems reasonable to assume that the proposed features can also be used to differentiate fallers from non-fallers. The proposed approach would also be valuable for regular follow-up of postural stability changes in older fallers in clinical practice so that appropriate assistance can be provided to reduce fall risk.

The proposed approach has some limitations. First, the cost of the force plates can limit the employment of the proposed approach. Second, since the effectiveness of the proposed features varies with the number of measurements, we still need to develop a strategy to determine the optimal number of measurements to balance the ease-of-use and the accuracy of the proposed approach. Third, by using state-of-the-art machine learning methods to combine the proposed features with other postural stability measures, it is possible that a better postural stability assessment method can be developed. This potential, however, has not been explored in this work.

## 5. Conclusions

Based on signals that can be readily obtained from a force plate, we introduced several postural stability features by quantifying the coupling strengths between the COP and its GRF components. The efficacy of the proposed features was demonstrated by comparing their performances in detecting the effects of aging on the quiet standing posture. Using statistical and binary classification tests, we showed that two of the proposed features outperformed one of the most effective conventional COP features.

Based on the values of the proposed features obtained from our experiments, we made the following observations, which elucidate the effect that the GRF components have on stabilizing the quiet standing posture. First, the COP was largely determined by the vertical GRF components in quiet standing. Second, despite the dominant role of the vertical GRF components, the contributions of the horizontal GRF components were not negligible. Our experimental results demonstrated that the AP-direction GRF component of the younger group played a more active role than that of the older group in positioning the COP. In fact, this difference was not only statistically significant but also very effective in differentiating postural performance between the younger group and the older group. Third, by having a weaker coupling strength with COP, the AP-direction GRF components of



the younger group were more flexible in maneuvering the motion of the COM and in controlling the FM. Considering the simplicity and efficacy of the proposed features, future studies can extensively test the effectiveness of our proposed features in characterizing impairments to postural stability from many health-related conditions.

**Author Contributions:** All authors participated in the research. Conceptualization, J.-L.S., C.-W.Y. and L.-J.L.; data curation, C.-H.L. and P.L.; formal analysis, J.-L.S., C.-W.Y. and L.-J.L.; investigation, L.-Y.G., P.L. and L.-J.L.; methodology, J.-L.S., L.-Y.G. and C.-W.Y.; Software, J.-L.S. and C.-H.L.; writing—original draft, C.-W.Y.; writing—review & editing, C.-W.Y. and L.-J.L. All authors discussed the results and contributed to the final manuscript. All authors have read and agreed to the published version of the manuscript.

**Funding:** Ministry of Science and Technology of Taiwan funded this research under the grant numbers MOST-106-2221-E-110-044 and MOST 107-2221-E-110-073.

**Acknowledgments:** This research was partly supported by the Ministry of Science and Technology in Taiwan (R. O. C.), under grants 106-2221-E-110-044 and MOST 107-2221-E-110-073.

**Conflicts of Interest:** The authors declare no conflict of interest.

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
