# Peer review of "Assessing Postural Stability Using Coupling Strengths between Center of Pressure and Its Ground Reaction Force Components"

_applsci, doi:10.3390/app10228077_

Round 1
Reviewer 1 Report
The authors investigate four postural stability features by quantifying the coupling strengths between the COP coordinates and their GRF components. Two small age groups are considered. The physical relationships between the GRF and the CoP are discussed in detail. This makes it easy for readers with technical background to understand the discussed relationships. However, the distinction to the analyses from the early period (1960-1980) of electronic posturography is not clear. This can also be seen in the literature that does not look back far enough to take up findings from this time. Unfortunately, this important topic has fallen out of focus of research in the last 30 years. It would be desirable if the authors could break down the goal of the study into a concise thesis. Likewise it would be desirable to present the benefit of the investigation in simple contexts so that the added value becomes clear for therapists and doctors. The discussion also lacks the connection to biotechnology. The abstract problems investigated here are ultimately products of the complex interplay of biomechanics and neuromuscular control.
Further remarks:
Methodology
Why was the measurement data filtered with a cut-off frequency of 5Hz? Is there a scientific validation for this? Studies on the filter cut-off frequency known to me explicitly advocate cut-off frequencies greater than 10Hz in order to eliminate non-essential useful components of the CoP signal.
data analysis
For a meaningful statistical evaluation, the effect size and power should always be specified for the anova. No valid conclusions could be drawn from the P value.
Which program was used to calculate the statistics?
Author Response
Please be advised that, in order to help the reviewer to keep track of the changes that we made, we have uploaded the PDF file of the revised manuscript.
Comment: The authors investigate four postural stability features by quantifying the coupling strengths between the COP coordinates and their GRF components. Two small age groups are considered. The physical relationships between the GRF and the CoP are discussed in detail. This makes it easy for readers with technical background to understand the discussed relationships. However, the distinction to the analyses from the early period (1960-1980) of electronic posturography is not clear. This can also be seen in the literature that does not look back far enough to take up findings from this time. Unfortunately, this important topic has fallen out of focus of research in the last 30 years. It would be desirable if the authors could break down the goal of the study into a concise thesis. Likewise it would be desirable to present the benefit of the investigation in simple contexts so that the added value becomes clear for therapists and doctors. The discussion also lacks the connection to biotechnology. The abstract problems investigated here are ultimately products of the complex interplay of biomechanics and neuromuscular control.
Response: We agree with the reviewer that it is unfortunate that the important topic of posturography has fallen out of focus of research in the last 30 years. We believe that one of the reasons for this unfortunate outcome is that, due to the limitations of conventional postural stability measures, researchers have difficulties to explore the potential of posturography. Addressing this problem is the goal of this study. By considering the interactions between COP and ground reaction force (GRF) components, we hope we can develop a different and useful direction to study posturography.
To help readers better understand the background and utility of posturography, the following two review papers have been added to the revised manuscript (lines 71-72 of the revised manuscript).
- Visser, J.E.; Carpenter, M.G.; Kooij, H.; Bastiaan B.R. The clinical utility of posturography. Clin Neurophysiol, 2008, 119(11), 2424-36. DOI:10.1016/j.clinph.2008.07.220
- Chaudhry, H.; Bukiet, B.; Ji, Z.; Findley, T. Measurement of balance in computer posturography: Comparison of methods--A brief review. J Bodyw Mov Ther. 2011, 15(1), 82-91. DOI:10.1016/j.jbmt.2008.03.003
To help therapists and doctors understand the added value of the proposed features, the following sentences have added to the end of the abstract.
Lines 47 – 49 of the revised manuscript: Based on these findings, it is believed that the proposed features can be used as a set of stability measures to assess the effects on posture stability from various health-related conditions such as aging and fall risk.
In addition, the following sentences have also been added to the revised manuscript.
Lines 344 – 353 of the revised manuscript: In specific, by performing longitudinal studies to observe how the values of the proposed features vary with the disease severity, the proposed approach can become an effective tool for monitoring the progress of diseases.
Perhaps one of the most important potential applications of the proposed approach is to identify older people with high fall risk. As addressed in a review paper, conventional COP features can be used to discriminate fallers from non-fallers [30]. Based on the favorable comparative results obtained in this work, it seems reasonable to assume that the proposed features can also be used to differentiate fallers from non-fallers. The proposed approach would also be valuable for regular follow-up of postural stability changes in older fallers in clinical practice so that appropriate assistance can be provided to reduce fall risk.
In responding to the excellent comment that problems investigated here are ultimately products of the complex interplay of biomechanics and neuromuscular control, we have added the following sentence to address this important insight.
Lines 339 – 341 of the revised manuscript: By quantifying the coupling strengths between COP and its GRF components, the proposed features try to characterize the complex interplay of biomechanics and neuromuscular control during the balancing process.
Further remarks:
Comment: Methodology
Why was the measurement data filtered with a cut-off frequency of 5Hz? Is there a scientific validation for this? Studies on the filter cut-off frequency known to me explicitly advocate cut-off frequencies greater than 10Hz in order to eliminate non-essential useful components of the CoP signal.
Response: One of the reasons why that COP signals have noised frequency components is that the ground reaction force (GRF) components measured by force plates have high frequency noised spectral components. This is the reason why this study used a low-pass filter to process the GRF signals. Regarding the cut-off frequency of such a low-pass filter, there does not seem to a universally accepted standard. We agree with the reviewer that many studies used 10 Hz or higher as the cut-off frequency. However, low pass filters with cut-off frequency setting at 5 Hz or lower have also been used (Prieto et al., 1996; Bottaro et al., 2008; Vette et al., 2010). In fact, Bottaro et al., (2008) explicitly stated that the frequency band of the COP does not exceed 3-5Hz. Our experimental data agree with their observation. The papers mentioned in this paragraph are listed as follows.
Prieto, T.E.; Myklebust, J.B.; Hoffmann, R.G.; Lovett, E.G.; Myklebust, B.M. Measures of postural steadiness: differences between healthy young and elderly adults. IEEE Trans Biomed Eng. 1996, 43(9), 956-66. DOI:10.1109/10.532130
Bottaro, A.; Yasutake, Y. Nomura, T.; Casadio, M.; Morasso, P. Bounded stability of the quiet standing posture: an intermittent control model. Hum Mov Sci. 2008, 27(3), 473-95. DOI:10.1016/j.humov.2007.11.005
Vette, A.H.; Masani, K.; Sin, V.; Popovic, M.R. Posturographic measures in healthy young adults during quiet sitting in comparison with quiet standing. Comparative Study Med Eng Phys. 2010, 32(1), 32-8. DOI:10.1016/j.medengphy.2009.10.005
Comment: data analysis
For a meaningful statistical evaluation, the effect size and power should always be specified for the anova. No valid conclusions could be drawn from the P value. Which program was used to calculate the statistics?
Response: This study used MATLAB to perform statistical tests. We agree with the reviewer that no valid conclusions can be drawn from the p value. This is the very reason why we also used the tested features to classify the two age groups. To address the problem of a limited number of participants, the revised manuscript has substantially revised the classification method by introducing a generalization test procedure to assess the effectiveness of the tested features. In specific, this generalization procedure randomly selected 80% of the participants (8 men and 8 women for each age group) as the training subset whereas the remaining participants (2 men and 2 women for each age group) were assigned to the testing subset. For each of the tested features, the decision threshold was determined from the training subset. This decision threshold was then used to classify the testing subset samples. With randomly divided training and testing subsets, this generalization test procedure was repeated 1,000 times for each of the tested features. The results of these generalization tests are summarized in Table 2. We believe that the average results of these 1,000 generalization test results can more reliably demonstrate the effectiveness of the tested features in discriminating the two age groups. As shown in Table 2, when samples were generated by the three-trial averaging technique, the classification accuracy of the testing subsets are relatively close to the classification accuracy of the training subset. This suggests that, even with a limited number of training subset participants, the proposed features still perform very well in dealing with testing subset participants whose information was completely hidden from the classification rule development process.
The revised manuscript has added the following sentences to illustrate the operating procedure of this newly introduced generalization test method.
Lines 212 – 224 of the revised manuscript: To assess the effectiveness of the tested features, one can use the entire dataset to determine the ROC curve and the corresponding optimal decision threshold. However, with a limited number of participants, whether such results can be accurately generalized to the general population is questionable. To address this issue, this study used a generalization test procedure to assess the effectiveness of the tested features. In specific, this generalization test procedure randomly selected 80% of the participants (8 men and 8 women for each age group) as the training subset and assigned the remaining 20% of the participants (2 men and 2 women for each age group) to the testing subset. For each of the tested features, this generalization procedure performed the following steps. First, based on the classification method described in the previous paragraph, this procedure used the training subset to determine the optimal decision threshold. Next, this decision threshold was used to classify the testing subset samples. With such a randomly divided training and testing subsets, this procedure was repeated 1,000 times for each of the tested features. To characterize the effectiveness of a tested feature, the final step determined the average accuracy of these 1,000 generalization test results.
The revised manuscript has also added the following sentences to discuss the results obtained by the generalization tests.
Lines 252– 272 of the revised manuscript: Note that, for each of the tested features, Table 2 provides three sets of classification results which include results of the training subset, three-trial averaging sample testing subset and, single- trail sample testing subset. To study the differences between these three sets of results, as a representative case, here we compare the results associated with feature βAP. As shown in Table 2, for βAP, the classification accuracy of the training subset, the three-trial averaging sample testing subset and, the single-trial sample testing subset are 0.903, 0.895 and, 0.839, respectively. As expected, the training subset has the best accuracy since the decision threshold was designed to minimize the classification error of the training subset samples. In comparison, by using such a decision threshold to classify the testing subset samples, the classification results are expected to be less accurate since the probability distribution function of the training subset is not identical to that of the testing subsets. Another important observation is that the classification accuracy of the training subset and the three-trial averaging sample testing subset are relatively close (0.903 versus 0.895). This suggests that, even with a relatively small number of training subset participants, βAP still performs very well in dealing with testing subset participants whose information was completely hidden from the classification rule development process.
For βAP, the testing subset classification accuracy of the three-trial averaging samples is better than that of the single-trial samples (0.895 versus 0.839). Such a difference is also expected since the decision threshold was determined by using three-trial averaging samples. Such a result also suggests that increasing the number of measurements can improve the effectiveness of βAP in differentiating the two age groups. Finally, although here we only explicitly discuss the results associated with βAP, the comparative results of the remaining tested features are basically the same.
Finally, we want to thank the reviewer for reviewing this paper. We also want to thank you for making many valuable comments. By addressing your comments, we hope that this work can be accepted for publication so that we can share our work with the research community as well as clinicians and therapists.

Reviewer 2 Report
The authors have hypothesized 4 metrics to assess postural stability. Each of these metrics are a mean of correlation coefficients for 2 traditional force plate metrics collected during a test of quiet standing posture. The 4 hypothesized metrics were compared to an established standard for assessing postural stability and seem to do a better job of detecting the subject’s age group. The manuscript is well-written with very few minor grammatical errors. The figure and equations are clearly presented. However, I have a few questions about some details of the approach. Clarifying these questions (see below) would greatly improve the manuscript for publication.
Line 126: Define the time subinterval and explain how this duration or number of subintervals within a trial was chosen.
Line 179: I’m unfamiliar with the use of a simple average technique to increase the number of samples beyond those datapoints collected. Please provide a reference verifying the accepted use of this method.
Line 196: I believe “negative positive rates” is a typo. Please correct or provide an explanation of this term.
Line 201: Separate Results and Discussion into 2 sections.
Line 254: I think you mean right-hand side of equation (13). If left-side, please explain.
Line 272: This final paragraph briefly touches on potential future uses of this method. Please expand the details of how this method can be used in the future. Also, please address any limitations of the study (i.e. data collection methods, subjects, analyses, etc.).
Discussion: While this method seems to differentiate between young and old subjects quite well, this doesn’t seem to be of much use in a clinical setting. I wonder whether this method could be used to inform differentiating whether a subject is at risk of falling or not, to inform whether assistance is needed. Please expand this idea how this method can be applied in real-world decision making in the future.
Author Response
Please be advised that, in order to help the reviewer to keep track of the changes that we made, we have uploaded the PDF file of the revised manuscript.
The authors have hypothesized 4 metrics to assess postural stability. Each of these metrics are a mean of correlation coefficients for 2 traditional force plate metrics collected during a test of quiet standing posture. The 4 hypothesized metrics were compared to an established standard for assessing postural stability and seem to do a better job of detecting the subject’s age group. The manuscript is well-written with very few minor grammatical errors. The figure and equations are clearly presented. However, I have a few questions about some details of the approach. Clarifying these questions (see below) would greatly improve the manuscript for publication.
Comment: Line 126: Define the time subinterval and explain how this duration or number of subintervals within a trial was chosen.
Response: In selecting the length of this time subinterval, we tested many different values. Via a trial-and-error process, the length of these non-overlapping subintervals was chosen to be 0.125 seconds. However, the results of the proposed features are not sensitive to this time subinterval length. To address this problem, the following sentence has been added to the revised manuscript.
Lines 136 – 137 of the revised manuscript: Via a trial-and-error process, the length of these non-overlapping subintervals was chosen to be 0.125 seconds.
Comment: Line 179: I’m unfamiliar with the use of a simple average technique to increase the number of samples beyond those datapoints collected. Please provide a reference verifying the accepted use of this method.
Response: We are not able to provide a reference for this simple average technique since we developed this technique by ourselves. The intention is to prevent overfitting problems caused by an insufficient number of samples. Because we have 12 trial results for each participant and the number of possible combinations for selecting three objects from 12 objects is 220, this three-trial averaging technique generated 220 samples for each participant. However, we also know that classifiers developed by this technique can not provide optimal classification results in dealing with single-trial samples.
To addressed this problem and the problem of a limited number of participants, based on your valuable comment and the suggestion of another reviewer, the revised manuscript has substantially revised the classification method by introducing a generalization test procedure to assess the effectiveness of the tested features in differencing two age groups. In specific, this generalization procedure randomly selected 80% of the participants (8 men and 8 women for each age group) as the training subset whereas the remaining participants (2 men and 2 women for each age group) were assigned to the testing subset. For each of the tested features, the decision threshold was determined from the training subset. This decision threshold was then used to classify the testing subset samples. With randomly divided training and testing subsets, this generalization test procedure was repeated 1,000 times for each of the tested features. We believe that the average results of these 1,000 generalization test results can more reliably demonstrate the effectiveness of the tested features. The results of these generalization tests are summarized in Table 2. As shown in Table 2, when samples were generated by the three-trial averaging technique, the classification accuracy of the testing subsets are relatively close to the accuracy of the training subset. This suggests that, even with a limited number of training subset participants, the proposed features still perform very well in dealing with testing subset participants whose information was completely hidden from the classification rule development process. It is believed that such results have successfully demonstrated the effectiveness of the tested features in discriminating the two age groups.
We are also aware that the performances of the classifiers developed by three-trial averaging samples are questionable in dealing with participants who have less than three trial results. To address this problem, the revised manuscript also used the proposed approach to classify single-trial results. The revised manuscript has added the following sentences to address this problem.
Lines 225 – 236 of the revised manuscript: One of the limitations of the three-trial averaging technique is that each of the participants needs to have at least three trial results. Therefore, the effectiveness of the proposed approach becomes unclear in dealing with subjects with less than three trial measurements. To address this problem, we repeated the generalization tests by revising the testing subset samples. In specific, the decision thresholds were still determined by using training samples generated from the three-trial averaging technique. However, the testing subset used the single-trial measurements of the testing subset participants to characterize the effectiveness of the tested features. Therefore, the number of samples associated with each of the testing subset participants reduced from 220 (the number of combinations of selecting 3 trails from 12 trials) to 12 (each participant has 12 trial results). Note that, by repeating this procedure 1,000 times with randomly divided training and testing subsets, this method tested 48,000 samples (12 trials/participants * 4 testing subset participants/test * 1000 generalization tests) for each age group.
The revised manuscript has added the following sentences to illustrate the operating procedure of this newly introduced generalization test method.
Lines 212 – 224 of the revised manuscript: To assess the effectiveness of the tested features, one can use the entire dataset to determine the ROC curve and the corresponding optimal decision threshold. However, with a limited number of participants, whether such results can be accurately generalized to the general population is questionable. To address this issue, this study used a generalization test procedure to assess the effectiveness of the tested features. In specific, this generalization test procedure randomly selected 80% of the participants (8 men and 8 women for each age group) as the training subset and assigned the remaining 20% of the participants (2 men and 2 women for each age group) to the testing subset. For each of the tested features, this generalization procedure performed the following steps. First, based on the classification method described in the previous paragraph, this procedure used the training subset to determine the optimal decision threshold. Next, this decision threshold was used to classify the testing subset samples. With such a randomly divided training and testing subsets, this procedure was repeated 1,000 times for each of the tested features. To characterize the effectiveness of a tested feature, the final step determined the average accuracy of these 1,000 generalization test results.
The revised manuscript has also added the following sentences to discuss the results obtained by the generalization tests.
Lines 252– 272 of the revised manuscript: Note that, for each of the tested features, Table 2 provides three sets of classification results which include results of the training subset, three-trial averaging sample testing subset and, single-trail sample testing subset. To study the differences between these three sets of results, as a representative case, here we compare the results associated with feature βAP. As shown in Table 2, for βAP, the classification accuracy of the training subset, the three-trial averaging sample testing subset and, the single-trial sample testing subset are 0.903, 0.895 and, 0.839, respectively. As expected, the training subset has the best accuracy since the decision threshold was designed to minimize the classification error of the training subset samples. In comparison, by using such a decision threshold to classify the testing subset samples, the classification results are expected to be less accurate since the probability distribution function of the training subset is not identical to that of the testing subsets. Another important observation is that the classification accuracy of the training subset and the three-trial averaging sample testing subset are relatively close (0.903 versus 0.895). This suggests that, even with a relatively small number of training subset participants, βAP still performs very well in dealing with testing subset participants whose information was completely hidden from the classification rule development process.
For βAP, the testing subset classification accuracy of the three-trial averaging samples is better than that of the single-trial samples (0.895 versus 0.839). Such a difference is also expected since the decision threshold was determined by using three-trial averaging samples. Such a result also suggests that increasing the number of measurements can improve the effectiveness of βAP in differentiating the two age groups. Finally, although here we only explicitly discuss the results associated with βAP, the comparative results of the remaining tested features are basically the same.
Comment: Line 196: I believe “negative positive rates” is a typo. Please correct or provide an explanation of this term.
Comment: This is indeed an error. It should be “true positive rate”. By varying the decision threshold to classify more samples into the positive class, some of the negative class samples can be incorrectly classified into the positive class. As a result, false positive rate and true positive rate vary simultaneously with the value of the decision threshold. The revised manuscript has corrected this error.
Comment: Line 201: Separate Results and Discussion into 2 sections.
Response: Based on this constructive comment, the revised manuscript has separated Results and Discuss into two sections.
Comment: Line 254: I think you mean right-hand side of equation (13). If left-side, please explain.
Response: We thank the reviewer for pointing out this error. The revised manuscript has corrected this error by changing “right-had side” to “left-hand side”
Comment: Line 272: This final paragraph briefly touches on potential future uses of this method. Please expand the details of how this method can be used in the future. Also, please address any limitations of the study (i.e. data collection methods, subjects, analyses, etc.).
Response: In responding to this constructive comment, the following sentences have been added to the revised manuscript to expand the details of how this method can be used in the future.
Lines 344 – 346 of the revised manuscript: In specific, by performing longitudinal studies to observe how the values of the proposed features vary with the disease severity, the proposed approach can become an effective tool for monitoring the progress of diseases.
In addition, the following paragraph has also been added to the revised manuscript to address the limitations of the proposed approach.
Lines 354 – 360 of the revised manuscript: The proposed approach has some limitations. First, the cost of the force plates can limit the employment of the proposed approach. Second, since the effectiveness of the proposed features varies with the number of measurements, we still need to develop a strategy to determine the optimal number of measurements to balance the ease-of-use and the accuracy of the proposed approach. Third, by using state-of-the-art machine learning methods to combine the proposed features with other postural stability measures, it is possible that a better postural stability assessment method can be developed. This potential, however, has not been explored in this work.
Comment: Discussion: While this method seems to differentiate between young and old subjects quite well, this doesn’t seem to be of much use in a clinical setting. I wonder whether this method could be used to inform differentiating whether a subject is at risk of falling or not, to inform whether assistance is needed. Please expand this idea how this method can be applied in real-world decision making in the future.
Response: Differentiating whether a subject is at risk of falls is indeed a very important problem. In responding to this comment, we have added the following sentences into the revised manuscript.
Lines 347 – 353 of the revised manuscript: Perhaps one of the most important potential applications of the proposed approach is to identify older people with high fall risk. As addressed in a review paper, conventional COP features can be used to discriminate fallers from non-fallers [30]. Based on the favorable comparative results obtained in this work, it seems reasonable to assume that the proposed features can also be used to differentiate fallers from non-fallers. The proposed approach would also be valuable for regular follow-up of postural stability changes in older fallers in clinical practice so that appropriate assistance can be provided to reduce fall risk.

Reviewer 3 Report
Please see attached word document for clearer reviewer comments
---------------------------------------------------------------------
Assessing postural stability using coupling strengths between center of pressure and its ground reaction force components - Review
Introduction:
Good, nice literature, succinct and purpose of investigation is clear
Methods:
Proposed features:
Nice diagram of force plate, clearly explained
Equations (7) and (8) could be explained more clearly:
Unless I was just having a slow day (maybe) it took me a moment to realise the reason for the polarity of the Fz components. I now realise that in (7) F1 and F4 are positive as that is the positive x side of the force plate and F2 and F3 are negative as that is the negative x side of the force plate. And in (8), F1 and F2 z are positive as that is the positive y side of the force plate and vice versa. Saying that, although I know the arrows in Figure 1. Point towards the positive sides of their respective axes perhaps this could be labelled more clearly (perhaps putting a +ve and –ve) on the associated sides of the arrows in the diagram.
Lines 113 – 115: Would it be clearer to say that COPx is determined by its vertical GRF component Fxz and its horizontal medial-lateral component fx and COPy is determined by its vertical GRF component Fyz and its horizontal anterior-posterior component fy. I know that in their respective frames that fx and fy are both horizontal components compared to the vertical ground reaction force however, in lines 134-136 the authors refer to fy being the GRF component in the horizontal direction for features aap BAP (anterior-posterior) and the remaining aML and BML being the mediolateral features using fx as the GRF component in the medial - lateral direction. Thus I think it could be clearer to the reader if the authors refer to fx and fy as the medial-lateral horizontal component and fy as the anterior posterior horizontal component throughout the paper.
Experimental procedure:
Good subject size (10M, 10F) for both a younger and older population. Tested control and eyes closed. Ignored first 5 and last 5 seconds of trial for consistency of data (once participants were in their natural steady state).
Line 157: One can assume that it was a low-pass Butterworth filter (with a 5 Hz cut off frequency – commonly used for human movement data) however the authors should state that it is a low pass filter for completeness.
Line 158-159: What was the reasoning behind 0.125 second time-intervals? Was it empirically chosen or just assumed to be a good time duration.
Data analyses:
Line 166: Good citations to justify reasoning.
Table 1. The results show that the anterior-posterior features show a significant difference between younger and older groups where-as ML do not. This is an interesting result! It shows that anterior-posterior postural stability becomes less prominent with age rather than medial-lateral postural stability. This may be useful in medical interventions and assessing postural change in patients (perhaps assessing risk of fall). Used a three-way averaging technique to increase training sample size of classifier. That is, chose the total number of combinations of three measurements in a total of 12 and averaged it, resulting in 220 measures x 20 persons. An effective way to increase the training size.
Results and discussion:
Table 2 reinforces the results in Table 1, showing that the anterior-posterior features are much more effective at classifying between the age groups than the mediolateral features. The results are convincing too, with BAP yielding a classification accuracy of 90% and AUC of 0.953.
The authors should make it clearer how the training and evaluation datasets were split. If the classifier is trained and tested on the same data then strong accuracies can be misleading due to overfitting and non-independency. Commonly, the data is randomly split at approximately 75%/25% for training and evaluation or a leave-one-out (LOO) method could be used. That is, train the data on a random selection of data from say 15 participants in the younger group and 15 participants in the older group and then evaluate it on the remaining 5 from each group. This means the classifier is trained and tested with total independency and the classification results are more valid. If the authors did not use these techniques, one should be implemented and the classification results recalculated as they could be misleading. If the authors did execute a training/evaluation data split then this should be clearly stated in the paper.
Also, to further evaluate classification capability, supervised machine learning models could be implemented. That is, train classification models such as support vector machines, random forests, multi-layered perceptron neural networks, logistic regression and evaluate their classification performance. This way, all the features are included in the same classifier and this may improve performance. I know one of the points of the paper is to assess which feature has the most importance in differentiating between age groups, however this information can be acquired using supervised machine learning models also by using methods to assess feature importance (https://towardsdatascience.com/feature-selection-techniques-for-classification-and-python-tips-for-their-application-10c0ddd7918b). This hyperlink shows some common feature ranking methods. PCA will not reflect the importance of each feature individually and so this technique will not be as useful. Perhaps these models can be implemented in the future studies proposed in line 276.
Conclusions:
Overall good conclusions and the results are beneficial to the wider-scientific community. The authors found that the anterior-posterior GRF component is an important feature for differentiating between younger and older groups. That, is it is more actively working in younger groups to maintain postural stability.
Key recommendations:
Make Figure 1. Clearer by highlighting positive and negative side of force plate channel axes.
State that the filter is low pass (if it is).
Clearly state how an evaluation/training data split was made for the classifier, if it was implemented. If a split was not made then the classifier would have been prone to overfitting and the classifier scores would be misleading. Thus if a split was not made then the authors should do this (using one of the recommended methods) and calculate new classification results (this is the only reason I suggested major revisions instead of minor - although this could be minor if it just was not stated properly).

Author Response
Please be advised that, in order to help the reviewer to keep track of the changes that we made, we have uploaded the PDF file of the revised manuscript.
Introduction:
Comment: Good, nice literature, succinct and purpose of investigation is clear
Response: First, we want to thank you for reviewing this paper. As will be illustrated in the following responses, we have substantially revised our manuscript based on your valuable comments.
Methods:
Proposed features:
Nice diagram of force plate, clearly explained
Comment: Equations (7) and (8) could be explained more clearly:
Response: To explain the roles of equations (7) and (8) more clearly, the following sentences have been added to the revised manuscript.
Lines 113 – 116: Equation (7) demonstrates how the four vertical GRF components are combined to influence the motion of COPx which is the ML direction coordinate of COP. Similarly, equation (8) shows how the four vertical GRF components are combined to change the motion of COPy which is the AP direction coordinate of the COP.
Comment: Unless I was just having a slow day (maybe) it took me a moment to realise the reason for the polarity of the Fz components. I now realise that in (7) F1 and F4 are positive as that is the positive x side of the force plate and F2 and F3 are negative as that is the negative x side of the force plate. And in (8), F1 and F2 z are positive as that is the positive y side of the force plate and vice versa. Saying that, although I know the arrows in Figure 1. Point towards the positive sides of their respective axes perhaps this could be labelled more clearly (perhaps putting a +ve and –ve) on the associated sides of the arrows in the diagram.
Response: We are sorry that we do not understand the meaning of +ve and -ve. However, to avoid the confusion of the coordinate system of Figure 1, the following sentences have been added to the revised manuscript.
Lines 95 – 96: Note that the arrows associated with the three axes point to the positive direction of the corresponding axes.
However, if the reviewer has any suggestions for clarifying the directions of the coordinate systems, we will be more than happy to make additional changes.
Comment: Lines 113 – 115: Would it be clearer to say that COPx is determined by its vertical GRF component Fxz and its horizontal medial-lateral component fx and COPy is determined by its vertical GRF component Fyz and its horizontal anterior-posterior component fy. I know that in their respective frames that fx and fy are both horizontal components compared to the vertical ground reaction force however, in lines 134-136 the authors refer to fy being the GRF component in the horizontal direction for features aap BAP (anterior-posterior) and the remaining aML and BML being the mediolateral features using fx as the GRF component in the medial - lateral direction. Thus I think it could be clearer to the reader if the authors refer to fx and fy as the medial-lateral horizontal component and fy as the anterior posterior horizontal component throughout the paper.
Response: This is an excellent suggestion. Throughout the paper, the revised manuscript now refers fx and fy as the medial-lateral horizontal component and fy as the anterior-posterior horizontal component, respectively.
Experimental procedure:
Good subject size (10M, 10F) for both a younger and older population. Tested control and eyes closed. Ignored first 5 and last 5 seconds of trial for consistency of data (once participants were in their natural steady state).
Comment: Line 157: One can assume that it was a low-pass Butterworth filter (with a 5 Hz cut off frequency – commonly used for human movement data) however the authors should state that it is a low pass filter for completeness.
Response: It was indeed a low-pass filter. We agree with the reviewer that we should explicitly identify this important property. Therefore, the revised manuscript has corrected this problem.
Line 158-159: What was the reasoning behind 0.125 second time-intervals? Was it empirically chosen or just assumed to be a good time duration.
Response: In selecting the length of this time subinterval, we tested many different values. Via a trial-and-error process, the length of these non-overlapping subintervals was chosen to be 0.125 seconds. However, the effectiveness of the proposed features is not sensitive to this time subinterval length. To illustrate how the time subinterval was specified, the following sentences have been added.
Lines 136– 137 of the revised manuscript: Via a trial-and-error process, the length of these non-overlapping subintervals was chosen to be 0.125 seconds.
Data analyses:
Comment: Line 166: Good citations to justify reasoning.
Response: We agree with the reviewer that references 37-41 provide valuable information about the tested COP features for assessing postural steadiness.
Comment: Table 1. The results show that the anterior-posterior features show a significant difference between younger and older groups where-as ML do not. This is an interesting result! It shows that anterior-posterior postural stability becomes less prominent with age rather than medial-lateral postural stability. This may be useful in medical interventions and assessing postural change in patients (perhaps assessing risk of fall). Used a three-way averaging technique to increase training sample size of classifier. That is, chose the total number of combinations of three measurements in a total of 12 and averaged it, resulting in 220 measures x 20 persons. An effective way to increase the training size.
Response: The three-trial averaging technique was introduced to increase the number of training samples so that we can obtain more reliable classification results. However, the effectiveness of the classifiers designed in this manner becomes unclear in dealing with participants with less than three trial results. As illustrated in the next response, by introducing a new generalization test procedure to assess the effectiveness of the tested features, the revised manuscript provides more information about the efficacy of this three-trial averaging technique.
Results and discussion:
Table 2 reinforces the results in Table 1, showing that the anterior-posterior features are much more effective at classifying between the age groups than the mediolateral features. The results are convincing too, with BAP yielding a classification accuracy of 90% and AUC of 0.953.
The authors should make it clearer how the training and evaluation datasets were split. If the classifier is trained and tested on the same data then strong accuracies can be misleading due to overfitting and non-independency. Commonly, the data is randomly split at approximately 75%/25% for training and evaluation or a leave-one-out (LOO) method could be used. That is, train the data on a random selection of data from say 15 participants in the younger group and 15 participants in the older group and then evaluate it on the remaining 5 from each group. This means the classifier is trained and tested with total independency and the classification results are more valid. If the authors did not use these techniques, one should be implemented and the classification results recalculated as they could be misleading. If the authors did execute a training/evaluation data split then this should be clearly stated in the paper.
Also, to further evaluate classification capability, supervised machine learning models could be implemented. That is, train classification models such as support vector machines, random forests, multi-layered perceptron neural networks, logistic regression and evaluate their classification performance. This way, all the features are included in the same classifier and this may improve performance. I know one of the points of the paper is to assess which feature has the most importance in differentiating between age groups, however this information can be acquired using supervised machine learning models also by using methods to assess feature importance (https://towardsdatascience.com/feature-selection-techniques-for-classification-and-python-tips-for-their-application-10c0ddd7918b). This hyperlink shows some common feature ranking methods. PCA will not reflect the importance of each feature individually and so this technique will not be as useful. Perhaps these models can be implemented in the future studies proposed in line 276.
Response: The reviewer made a very strong point in indicating that the effectiveness of the tested features should be more rigorously and comprehensively tested by splitting the dataset into training and evaluation subsets. Therefore, we have substantially revised the classification method to address this important problem.
In specific, the revised manuscript introduces a generalization test procedure to assess the effectiveness of the tested features in differencing two age groups. This generalization procedure randomly selected 80% of the participants (8 men and 8 women for each age group) as the training subset whereas the remaining participants (2 men and 2 women for each age group) were assigned to the testing subset. For each of the tested features, the decision threshold was determined from the training subset. This decision threshold was then used to classify the testing subset samples. With randomly divided training and testing subsets, this generalization test procedure was repeated 1,000 times for each of the tested features. We believe that the average results of these 1,000 generalization test results can more reliably demonstrate the effectiveness of the tested features. The results of these generalization tests are summarized in Table 2. As shown in Table 2, when samples were generated by the three-trial averaging technique, the classification accuracy of the testing subsets are relatively close to the classification accuracy of the training subset. This suggests that, even with a limited number of training subset participants, the proposed features still perform very well in differentiating testing subset participants whose information was completely hidden from the classification rule development process. It is believed that such results have successfully demonstrated the effectiveness of the tested features in discriminating the two age groups.
The revised manuscript has added the following sentences to illustrate the operating procedure of this newly introduced generalization test method.
Lines 212 – 224 of the revised manuscript: To assess the effectiveness of the tested features, one can use the entire dataset to determine the ROC curve and the corresponding optimal decision threshold. However, with a limited number of participants, whether such results can be accurately generalized to the general population is questionable. To address this issue, this study used a generalization test procedure to assess the effectiveness of the tested features. In specific, this generalization test procedure randomly selected 80% of the participants (8 men and 8 women for each age group) as the training subset and assigned the remaining 20% of the participants (2 men and 2 women for each age group) to the testing subset. For each of the tested features, this generalization procedure performed the following steps. First, based on the classification method described in the previous paragraph, this procedure used the training subset to determine the optimal decision threshold. Next, this decision threshold was used to classify the testing subset samples. With such a randomly divided training and testing subsets, this procedure was repeated 1,000 times for each of the tested features. To characterize the effectiveness of a tested feature, the final step determined the average accuracy of these 1,000 generalization test results.
The revised manuscript has also added the following sentences to discuss the results obtained by the generalization tests.
Lines 252– 272 of the revised manuscript: Note that, for each of the tested features, Table 2 provides three sets of classification results which include results of the training subset, three-trial averaging sample testing subset and, single- trail sample testing subset. To study the differences between these three sets of results, as a representative case, here we compare the results associated with feature βAP. As shown in Table 2, for βAP, the classification accuracy of the training subset, the three-trial averaging sample testing subset and, the single-trial sample testing subset are 0.903, 0.895 and, 0.839, respectively. As expected, the training subset has the best accuracy since the decision threshold was designed to minimize the classification error of the training subset samples. In comparison, by using such a decision threshold to classify the testing subset samples, the classification results are expected to be less accurate since the probability distribution function of the training subset is not identical to that of the testing subsets. Another important observation is that the classification accuracy of the training subset and the three-trial averaging sample testing subset are relatively close (0.903 versus 0.895). This suggests that, even with a relatively small number of training subset participants, βAP still performs very well in dealing with testing subset participants whose information was completely hidden from the classification rule development process.
For βAP, the testing subset classification accuracy of the three-trial averaging samples is better than that of the single-trial samples (0.895 versus 0.839). Such a difference is also expected since the decision threshold was determined by using three-trial averaging samples. Such a result also suggests that increasing the number of measurements can improve the effectiveness of βAP in differentiating the two age groups. Finally, although here we only explicitly discuss the results associated with βAP, the comparative results of the remaining tested features are basically the same.
.
We also agree with the reviewer that by using more advanced machine learning methods such as multi-layered perceptron neural networks and XGBoost algorithm, we can combine the tested features to classify the two age groups. We also want to thank the reviewer for providing a very informative resource (https://towardsdatascience.com/feature-selection-techniques-for-classification-and-python-tips-for-their-application-10c0ddd7918b) for feature selection and ranking. However, at this stage, we simply do not have enough time to complete this task. This is indeed a limitation of this study. To address this limitation, the following sentences have been added to the revised manuscript to address this problem as well as other limitations of this work.
Lines 354 – 360: The proposed approach has some limitations. First, the cost of the force plates can limit the employment of the proposed approach. Second, since the effectiveness of the proposed features varies with the number of measurements, we still need to develop a strategy to determine the optimal number of measurements to balance the ease-of-use and the accuracy of the proposed approach. Third, by using state-of-the-art machine learning methods to combine the proposed features with other postural stability measures, it is possible that a better postural stability assessment method can be developed. This potential, however, has not been explored in this work.
Conclusions:
Overall good conclusions and the results are beneficial to the wider-scientific community. The authors found that the anterior-posterior GRF component is an important feature for differentiating between younger and older groups. That, is it is more actively working in younger groups to maintain postural stability.
Key recommendations:
Comments: Make Figure 1. Clearer by highlighting positive and negative side of force plate channel axes.
Response: The following sentences have been added to the revised manuscript to clarify this problem.
Lines 95 – 96: Note that the arrows associated with the three axes point to the positive direction of the corresponding axes.
However, if the reviewer has any suggestions for clarifying the directions of the coordinate systems, we will be more than happy to make additional changes.
Comment: State that the filter is low pass (if it is).
Response: The revised manuscript explicitly identifies our filter as the low-pass filter.
Comment: Clearly state how an evaluation/training data split was made for the classifier, if it was implemented. If a split was not made then the classifier would have been prone to overfitting and the classifier scores would be misleading. Thus if a split was not made then the authors should do this (using one of the recommended methods) and calculate new classification results (this is the only reason I suggested major revisions instead of minor - although this could be minor if it just was not stated properly).
Response: In responding to the comment made for the Results and Discussion section, we have illustrated how the revised manuscript addresses this important problem. The previous manuscript did not split the dataset into training and testing subsets. As a result, it was impossible to demonstrate to generalization capability of the proposed approach. However, by randomly dividing the dataset into training and testing subsets and used the decision rule developed by the training subset to classify the samples of the testing subset, the revised manuscript rigorously demonstrates the generalization capability of the proposed approach by repeating the generalization tests 1,000 times with randomly divided training and testing subsets. As shown by the results of Table 2, the classification accuracy of the training and testing subsets are quite close when samples from both subsets were generated by using the three-trail averaging technique. These results suggest that, even with a small number of training subset participants, the proposed approach can still very reliably differentiate testing subset participants whose information was completely hidden from the classification decision development process. We hope that the reviewer agrees that this major revision has successfully addressed the concern that overfitting may lead to misleading results.

Round 2
Reviewer 1 Report
The changes of the authors are satisfactory and sufficient. in this state a publication can be agreed.
Reviewer 3 Report
The authors addressed my concerns very well. I believe the paper should now be published as it adds value to the scientific community.